# Adaptation Mechanisms of Small Ruminants to Environmental Heat Stress

**DOI:** 10.3390/ani9030075

**Published:** 2019-02-28

**Authors:** Haile Berihulay, Adam Abied, Xiaohong He, Lin Jiang, Yuehui Ma

**Affiliations:** 1Institute of Animal Science, Chinese Academy of Agricultural Sciences (CAAS), Beijing 100193, China; haile.berihulay@yahoo.com (H.B.); aa.abied89@gmail.com (A.A.); hexiaohong@caas.cn (X.H.); jianglin@caas.cn (L.J.); 2The Key Laboratory for Farm Animal Genetic Resources and Utilization of Ministry of Agriculture of China, Institute of Animal Science Chinese Academy of Agricultural Sciences (CAAS), Beijing 100193, China

**Keywords:** adaptation, behavioral, morphological, heat stress, physiological, small ruminant

## Abstract

**Simple Summary:**

Heat stress is an intriguing factor that negatively influences livestock production and reproduction performance. Sheep and goat are among the livestock that can adapt to environmental heat stress via a combination of physiological, morphological, behavioral, and genetic bases. Sheep and goat are able to minimize adverse effect of high thermal stress by invoking behavioral responses such as feeding, water intake, shade seeking, and increased frequency of drinking. Their morphological mechanisms are comprised of body shape and size, light hair color, lightly pigmented skin, and less subcutaneous fat, and the physiological means are that of increased respiration rate (RR), increased sweating rate (SW), reduced metabolic rate, and change in endocrine function. Adaptation in terms of genetics is the heritable trait of animal characteristics which favor the survival of populations. For instance, genes like heat shock proteins 70 (HSP70) and *ENOX2* are commonly expressed proteins which protect animals against heat stress.

**Abstract:**

Small ruminants are the critical source of livelihood for rural people to the development of sustainable and environmentally sound production systems. They provided a source of meat, milk, skin, and fiber. The several contributions of small ruminants to the economy of millions of rural people are however being challenged by extreme heat stress difficulties. Heat stress is one of the most detrimental factors contributing to reduced growth, production, reproduction performance, milk quantity and quality, as well as natural immunity, making animals more vulnerable to diseases and even death. However, small ruminants have successfully adapted to this extreme environment and possess some unique adaptive traits due to behavioral, morphological, physiological, and largely genetic bases. This review paper, therefore, aims to provide an integrative explanation of small ruminant adaptation to heat stress and address some responsible candidate genes in adapting to thermal-stressed environments.

## 1. Introduction

Small ruminants (sheep and goat) are a critical source of livelihood for rural people to the development of sustainable and environmentally sound production systems, mainly in the extreme heat-stressed environments. Heat stress is one of the several distraction factors that make animal production and reproduction difficult in many areas of the world. It results in impaired production, reproduction, growth, milk quantity and quality, as well as natural immunity and making the animals more susceptible to different diseases [1]. Heat stress affects ruminant animals by the combination of environmental factors (high ambient temperature, relative humidity, high solar radiation, low wind speed, and precipitation) [2]. Air temperature and relative humidity are mainly direct influences on ruminant animal production potentials [3]. However, in extreme environmental conditions, sheep and goat perform better heat stress additivity than other ruminant animals [2,4]. They are the most adaptable and geographically widespread livestock species, ranging from the high mountains of hypoxia to the extreme low land of a thermal stressed environment [5,6]. 

Sheep and goats adapt to extreme weather conditions via behavioral, morphological, physiological, and largely genetic bases [3,7]. Morphologically, the coat color plays an important role in the evolved adaptation. Animals with light coat coloring absorb less heat than those with darker coats [8]. There is also a number of evidence that indicates adaptive genetic variations that enable living in heat-stressed environments, for example in sheep [3,9], goat [10,11,12], and both sheep and goat [13]. 

Investigation on genes/candidate genes associated with heat tolerance adaptability that have undergone natural selection in low-altitude dweller species is a key concern for researchers. For instance, genes like heat shock proteins 70 (HSP70) are commonly expressed proteins that protect animals against heat stress [10]. Further, the *ENOX2* gene is highly expressed in heat-stress-susceptible and -stress-tolerant goat. Similarly, the *MCIR*, *ASIP,* and *TYRP1* genes are highly polymorphic genes observed in locally adapted sheep breeds that are associated with wool color [14]. 

In brief, the adaptation is the level of tolerance to survive and reproduce under extreme living conditions [15]. Sheep and goats are very rustic animals that can cope with such an environment. However, full information on how these animals can adapt and survive to the novel and transforming environments are lacking. Accordingly, the underlying adaptation mechanisms and the specific genes involved in such areas are vital to investigate future selection programs of these animals. Therefore, the purpose of this review paper is (1) to provide an integrative explanation of the sheep and goat adaptation mechanisms in heat-stressed environments, and (2) to assess the specific genes and candidate genes involved in heat tolerance.

## 2. Small Ruminant Adaptation Mechanisms to Heat Stress

Nowadays, heat stress is a significant concern in the ever-changing climatic scenario. It is quite known that heat stress affects ruminant animals, particularly through decreased reproduction, growth, and production, increased health issues and mortality [1]. However, sheep and goat are less susceptible to heat-stressed environment than other ruminant animals [4]. Behavioral, morphological, physiological, and genetic bases are among the key adaptation mechanisms of small ruminants that respond in heat-stressed environments [15]. 

### 2.1. Morphological Adaptation

Morphological adaptations are physical changes that occur over many generations of animals that enhance its fitness in a given environment. Body size and shape, coat and skin color, hair type, and fat storage are among the main morphological adaptation in sheep and goat [16]. Table 1 shows morphological adaptations of sheep and goat.

#### 2.1.1. Body Size and Shape

Body size and shape are the most dominant morphological characteristics influencing the thermoregulatory mechanisms of farm animals in extremely hot environments [22]. Naturally, animals are characterized with large or small or dwarf bodies, which helps them to adjust water loss and heat gain in extremely hot environments [23]. The authors stated that animals with larger body size have lower metabolic rate than that of smaller animals and gain heat at a slower rate. The rate of heat gain and loss in an animal is highly influenced by body shape and appendages; for instance, animals with long and narrow appendages reduce the radiant of heat gain but increase convective heat loss. According to Ref. [24] taller animals dissipate more heat than animals with short, squat bodies. The Bighorn sheep have a compact muzzle; narrow, pointed, short ears; a short tail; curled back horns over the ear; deer-like fur; and are usually brown shade with whitish rump, and they are a well-adapted breed living in desert-hot environments [25]. Similarly, Sudanese and Egyptian desert goats have relatively large to medium body size (33 kg), which helps for evaporative heat loss [19]. Ethical approval on animal survival was given by the animal ethics committee of IAS-CAAS with the following reference number: IASCAAS-AE-03. 

#### 2.1.2. Coat and Skin Color

Coat and skin colors are an important trait of biological, economic, and social significance in animals. The coat and skin color characteristics of sheep and goat that have evolved in tropical and desert areas are different from those that evolved in temperate climates. For instance, the loose, open fleece of hair and wool of Awassi sheep enhances heat loss via convection [17]. Coat color is the simplest characteristic to look for in identifying the breed of sheep and goat population since it is easily and quickly observed. It is an important feature of the sheep and goat for determining the radiant heat load and how much solar radiation is reflected from their body and how much is absorbed [26]. Animals with light coat coloring absorb less heat than those with darker coats [8]. For example, West African dwarf goats have smooth, short, and straight hair, which helps them adapt in hot, humid environments [26]. According to Ref. [27], sheep with dark pigmentation are more prone to heat stress than those with light pigmentation, suggesting that selection should target animals with light coat color in order to improve animal welfare and production efficiency. The authors also revealed that rectal temperature (RT), respiration rate (RR), pulse rate (PR), packed red cell volume (PRCV), plasma sodium (Na^+^), and potassium (K^+^) are effects of coat color associated with climatic-stress-tolerance traits in four sheep color categories of West African dwarf sheep. Besides this, Awassi [17] and Omani [28] sheep are among the carpet-type sheep breeds. Sheep with carpet-type wool, light-colored fleece, thinner skin, shorter hairs, and fatter tails have better heat dissipation in hot environments [29]. The brown and black Santa Ines ewes have thicker skin and longer hair and are less adapted to high ambient temperatures compared to white ewes, which reflects lower rectal temperatures and respiratory rates at higher ambient temperatures [29].

#### 2.1.3. Fat Storage

Most of the time, fat tails and rump fat are considered as an adaptive response of animals to extreme environments and are used as a valuable energy reserve for the animal during migration and winter [30]. About 25% of the world’s sheep population comprises fat-tailed breeds [30]. Breeds such as Pelibuey sheep of Mexico are the most extensively employed maternal breeds in the tropical conditions and have an ability for accumulation and mobilization of body fat in the internal fat depots [31]. Besides this, the barbarian Tunisian sheep is a fat-tailed breed and they have the ability to deposit and mobilize body reserves not only from the tail (fat) but also from the rest of the body (i.e., body fat and body protein) [32]. 

### 2.2. Behavioral Adaptation

Animals behave in various ways during heat stress, and that can provide insight on how and when to cool them [15]. For instance, ruminants are active during the day and rest during the night. They protect themselves from extreme environmental factors by means of dissipating body heat by taking advantage of hairlessness of certain body parts, shedding of hair, water restriction, and feed intake [33,34]. 

When animals are exposed to high temperatures, reduction of feed intake will occur [33]. Reducing feed intake is a method of adaptation to decrease heat production in the warm environment as the heat increment of feeding is an important source of heat production in ruminants [33]. Goats are better adapted to heat stress than cows and sheep. They have a dynamic eating behavior during warm weather conditions. For example, Fawn goats have different eating behaviors in comparison with Saanen x hair goats, which are exposed to severe heat stress and poor nutritional condition [34]. The authors indicated that Saanen goats had higher meal size and meal length and longer inter-meal interval, meal time, and eating rate within each meal, but a lower number of meals in comparison with German Improved Fawn (GIF) goats. Goats exposed to heat-stressed environments are decreased in feed intake, body weight, and growth rate [33]. Thermal heat index (THI) is a good indicator of the stressful thermal climatic condition [34]. According to Ref. [34], THI 70 or lower values are considered as comfortable, 75–78 is stressful, and values higher than 78 are distressful because animals are unable to maintain thermoregulatory mechanisms or normal body temperature. 

### 2.3. Physiological Mechanisms

Animals possess a variety of physiological adaptation mechanisms that help in the reduction of heat load [1]. When the physiological mechanism fails to alleviate the effect of heat load, the body temperature may increase to a point at which animal well-being is compromised. Body temperature is a good measure of heat tolerance in animals, as it represents the result of all heat gain and heat loss processes in the body. Change in heart rate (HR), respiration rate (RR), and rectal temperature (RT) are the key parameters that indicate the mechanism of physiological adaptation in small ruminants [10,35]. Rectal temperature is a good index of body temperature even though there is a considerable variation in several parts of the body score at different times of the day. In heat-stressed environments, respiratory rate is the first thermoregulation mechanism used by sheep and goat to help them maintain their body temperature. Furthermore, panting is another physiological mean recognized as sheep’s response to increased environmental heat through a substantial increase in respiratory rate [15]. It is important to discuss that complexity and a suite of physiological changes due to heat stress response can differ from one animal to other and from individual to individual. When animals exposed to the high ambient temperature of about 40, 42, and 44 °C, the respiration rate, pulse rate, and rectal temperature increase [36]. For example, Morada Nov sheep have a lower respiratory rate than Santa Ines, which indicates better adaptation to high ambient temperature [3]. 

Physiologically, sheep and goat thrive in heat-stressed environment through increased respiration rate, increased sweating rate, changes in endocrine function, and reduced metabolic rate [37]. When the physiological mechanism fails to alleviate the effect of heat load, the body temperature may increase to a point at which animal well-being is compromised. Different literature has shown that respiratory rate and rectal temperature are good indicators of the thermal stress and may be used to assess the adversity of the thermal environment. For instance, [38] revealed that there was an increase in rectal temperature and respiration rate from 38.97 and 43.66 °C to 39.35 and 77.33 °C, respectively, when goats were kept for 6 hours in hot ambient temperature. Similarly, the study by Ref. [36] reported increases in RR and RT from 40, 42, and 44 °C, as the animal was exposed to higher temperatures. The authors stated that the proportion of increase in RR became higher with the increase in exposure temperature and RR reached up to 98.28 breaths per minute at 44 °C. Respiration rate increased significantly at temperature 40 °C but further increase at 42 and 44 °C was not significant [36]. Consistent with this study, significant increases in RR and RT in ewes subjected to walking stress by allowing them to walk for 14 km a day have been reported by Ref. [37]. As temperature increases above the thermal comfort zone, the rate of sensible heat loss gets reduced, and evaporative cooling mechanisms are activated. Heat-stressed sheep are found to primarily depend on respiratory and cutaneous means of cooling adjustments [39].

### 2.4. Genetic Bases of Adaptation

Adaptation in terms of genetics refers to the heritable traits of animal characteristics that favor the survival of populations [40]. Adaptation traits are usually characterized by low heritability. Every animal that has successfully adapted to a certain habitat possesses some unique adaptive traits that may be due to behavioral, morphological, physiological as well as genetic bases [1]. Genetic variation in a population provides flexibility to adapt to the changing environment and it is crucial for the survival of the population over time. The genetic basis of heat adaptation is poorly understood. Evidence from different researchers indicated that the role of genetics in determining an individual’s capability to adapt to the stressed environment is very complicated. As a result, in the cellular energy, mitochondria play a central role as a facilitator of energy metabolism [41]. 

The mitochondrial genes are highly associated genes in adaptability [42]. The organelles contain their own genome with the modified genetic code. Some studies suggest that, in animals, the mitochondrial genome is a circular, double-stranded molecule with a length of about 15–17 kb [41], is found in all eukaryotic cells, and is known for its function in energy supply [43,44]. Fast evolutionary rate, relatively conserved gene content and organization, small size, maternal inheritance, and limited recombination are among the sole characteristics of mitochondrial DNA (mtDNA) [45].

## 3. Genes and Candidate Genes Associated with Heat-Stressed Environments

Evolution is a continuous process that molds and remolds combination of genes in living things to adapt to the changing environment. Genome and genomic studies help to investigate thermo-tolerance genes and genomic regions that play a significant role for regulation of body temperature in small ruminants [13]. Animal adaptation to hot arid environments may be mediated by a complex network of genes [46].

A study by Ref. [47] reveals that an approach using genome-wide DNA markers improves genomic selection tolerance to heat stress and other traits. Similarly, [13] detected candidate genes associated in Egypt Baraki desert sheep and goat. The authors found sheep chromosome 10 (OAR10), which spanned several genes associated with stress, e.g., tumor suppressors, angiogenesis, and wound healing, whereas in goat, genes associated with stress were located in a selection sweep region (26–46 Mb) that were found on chromosome 6. Furthermore, the study by Ref. [9] indicated that *GPX3* is located in the arachidonic acid metabolism pathway and it is important for the survival of sheep in desert environments. Besides to this, Ref. [46] also reported 11 candidate selection sweep regions across 12 chromosomes in Egypt sheep in the hot arid environment. Table 2 indicates genes/candidate genes involved in heat tolerance.

## 4. Conclusions

This review highlights adaptation mechanisms of small ruminants in heat-stressed environments. Heat stress happens at the point where the animals cannot dissipate an adequate quantity of heat to maintain body thermal balance. Among them, high temperature, high humidity, and radiant energy are the major environmental factors that contribute to heat stress in animals. Animals such as sheep and goat adapt to heat-stressed weather conditions via a combination of behavioral, morphological, physiological as well as genetic bases. Furthermore, the review addresses specific genes and candidate genes that play a significant role in protecting against heat stress. However, detailed investigations have to be done regarding the adaptation of small ruminants in local stressed weather conditions.

## Figures and Tables

**Table 1 animals-09-00075-t001:** The key morphological adaptations in sheep and goat.

Key Morphological Characteristics	Animal	Reference
Loose coarse wool and Adipose tissue reserves	Awassi sheep	[17]
Fat-tail	Damara sheep	[18]
Long legs and long ears	Sudan Sahel and Egyptian Zaraiby goats	[19]
Short-legged	West African goat	[20]
Coat color	Massese, Xalda and Soay sheep	[21]
Skin pigmentation	Barki sheep and goat	[13]

**Table 2 animals-09-00075-t002:** Genes/candidate genes involved in heat tolerance.

Candidate Genes	Function	Breeds	Reference
*ANXA6*, *GPX3*, *GPX7,* and *PTGS2*	Arachidonic acid metabolism	Taklimakan desert sheep	[9]
*CPA3*, *CPVL*, and *ECE1*	Renin–angiotensin system	Taklimakan desert sheep	[9]
*CALM2*, *CACNA2D1*, *KCNJ5,* and *COX2*	Oxytocin signaling	Taklimakan desert sheep	[9]
*RAP1A*, *SLC4A4*, *CPA3,* and *CPB1*	Pancreatic secretion	Taklimakan desert sheep	[9]
*OXT, AVP, MICU2, IFT88*	Regulation of homeostatic process, Reproductive physiology and response to nutrient levels/digestive system	Baraki Sheep	[13]
*UROD*	Pigment biosynthetic process	Baraki Sheep	[13]
*EIF2B3*	Heat stress/temperature stimuli	Baraki Sheep	[13]
*PLK3*	Osmotic stress	Baraki Sheep	[13]
*TGM3*	Hair follicle morphogenesis	Baraki Sheep	[13]
*MCIR*, *ASIP*, and *TYRP1*	Coat color pattern	Crioula sheep	[14]
*HSP-70*	Protect cell against thermal injury	Pelibuey and Suffolk sheep	[48]
*HSP-70*	Thermotolerant	Mexico goat	[49]
*ASIP* *, KITLG, HTT, GNA11,and OSTM1*	Coloration	Chines goat	[12]
*TBX15* *, DGCR8, CDC25A, and RDH16*	Body size	Chines goat	[12]
*FGF2, GNAI3, PLCB1*	Thermo-tolerance (melanogenesis)	Baraki Sheep and goat	[13]
*BMP2, BMP4, GJA3, GJB2*	Body size and development	Baraki Sheep and goat	[13]
*MYH, TRHDE, ALDH1A3*	Energy and digestive metabolism	Baraki Sheep and goat	[13]
*GRIA1, IL2, IL7, IL21, IL1R1*	Nervous and autoimmune response	Baraki Sheep and goat	[13]
*TRPM8*	Regulation of body temperature	Brazilian sheep	[50]
*IL10RB and IL23A*	Immune response	Ugandan goat	[51]

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
