# Peer review of "Adaptation Mechanisms of Small Ruminants to Environmental Heat Stress"

_animals, 2019, doi:10.3390/ani9030075_

Reviewer 1 Report

The title should be "adaptation mechanisms of small ruminants to environmental heat stress

The english should be improved 

I do not understand the acknowledgement section this is only review study, how any foundation could support?

Author Response

Response to Reviewer 1:

The title should be "adaptation mechanisms of small ruminants to environmental heat stress

Response:

Thank you for your comments, we have corrected sentence accordingly

The English should be improved 

Response:

We thank very much the reviewer for the substantially comments and we carefully revise the grammar of the manuscript

I do not understand the acknowledgement section this is only review study, how any foundation could support?

Response:

We thank the reviewer for the suggestion; we have revised acknowledgement section accordingly

Reviewer 2 Report

General considerations - This review presents an interesting but not really original study on physiological adaptation mechanisms of small ruminants to heat stress environment, by taking into account the behavioural, morphological and physiological bases.

The language is appropriate and overall the manuscript is organized, nevertheless many grammar mistakes are present. The Abstract summarizes well the objectives and results of the review.   The Introduction is concise, well written and relevant to the subject. The keywords could be well designed (Behavioral adaptation; Morphological basis; Physiological mechanisms). The Tables are more detailed and well displayed, References are adequate and specific.

The review itself is interesting and the results are clearly useful and predictable.

I put some comments for further revisions and my considerations are described below:

Introduction: lines 35,  40, 66. Delete "small ruminants" and bracket.

Lines 41, 68,186, 223: change "basis" with "bases".

Line 47: delete "of".

Line 72: change "its" with "their".

Lines 81, 207: change "author" with "authors".

Line 85: change reduces with reduce.

Lines 87, 104: change "has" with "have".

Line 125: change "that" with "this".

Line 127: change "protecting" with "protect".

Line 135: delete the first "exposed".

Line 136: change "S" with "Saanen".

Line 138: please, clarify the term "GIF".

Line 149: add the acronym in the first term: heart rate (HR), respiration rate (RR), rectal temperature (RT) and put them in overall text.

Line 152: change "in different part" with "in several parts".

Line 1896: change "has been "with "have been".

Line 187: put the subject: "it is crucial for....."

Line193: change "suggests" with "suggest".

Line 205: change "improve" with "improves".

Conclusion: lines 218, 225. Change "ruminant" with "ruminants"

Lines 159, 220: change ";" with ","

In my opinion, this review, while important in subject, is acceptable for publication after minor  revisions.

Author Response

Thank you for your comments, which have helped us to substantially improve the manuscript. We have revised our manuscript in response to the given suggestions and questions as below. Changes to the manuscript are shown in red color.

Abstract:

1.      Introduction: lines 35, 40, and 66. Delete "small ruminants" and bracket.

Response: As suggested by the reviewer, we have delated based on the given comment

2.      Lines 41, 68,186, 223: change "basis" with "bases".

        Response:The suggested correction has been made

3.      Line 47: delete "of".

Response:

We thank the reviewer; we have delated

4.      Line 72: change "its" with "their".

       Response:

We thank the reviewer; we have changed

5.      Lines 81, 207: change "author" with "authors".

Response:

We thank the reviewer; we have changed

6.      Line 85: change reduces with reduce.

Response:

We thank the reviewer; we have changed

7.      Lines 87, 104: change "has" with "have".

Response:

We thank the reviewer; we have changed

8.      Line 125: change "that" with "this".

Response:

We thank the reviewer; we have changed

9.      Line 127: change "protecting" with "protect".

Response:

We thank the reviewer; we have changed

10.  Line 135: delete the first "exposed".

Response:

We thank the reviewer; we have delated as suggested

11.  Line 136: change "S" with "Saanen".

       Response:

We thank the reviewer; we have changed

12.  Line 138: please, clarify the term "GIF".

Response:

We thank the reviewer; we have clarify it

13.  Line 149: add the acronym in the first term: heart rate (HR), respiration rate (RR), rectal temperature (RT) and put them in overall text.

Response:

We thank the reviewer; we have add the acronym in the first term

14.  Line 152: change "in different part" with "in several parts".

Response:

We thank the reviewer; we have changed

15.  Line 1896: change "has been "with "have been".

Response:

We thank the reviewer; we have changed

16.  Line 187: put the subject: "it is crucial for....."

        Response:

We thank the reviewer; we have changed

17.  Line193: change "suggests" with "suggest".

Response:

We thank the reviewer; we have changed

18.  Line 205: change "improve" with "improves".

Response:

We thank the reviewer; we have changed

19.  Conclusion: lines 218, 225. Change "ruminant" with "ruminants"

Response:

We thank the reviewer; we have changed

20.  Lines 159, 220: change ";" with ","

Response:

We thank the reviewer; we have changed
